# Delivery of Foreign Materials into Adherent Cells by Gold Nanoparticle-Mediated Photoporation

**DOI:** 10.3390/membranes11080550

**Published:** 2021-07-22

**Authors:** Xiaofan Du, Jing Wang, Lan Chen, Zhenxi Zhang, Cuiping Yao

**Affiliations:** Key Laboratory of Biomedical Information Engineering of Ministry of Education, Institute of Biomedical Photonics and Sensing, School of Life Science and Technology, Xi’an Jiaotong University, Xi’an 710049, China; xiaofandu001@163.com (X.D.); wangjing@mail.xjtu.edu.cn (J.W.); chenlanship@xjtu.edu.cn (L.C.); zxzhang@mail.xjtu.edu.cn (Z.Z.)

**Keywords:** photoporation, adherent cells, effective delivery method, focused nanosecond laser

## Abstract

Delivering extracellular materials into adherent cells presents several challenges. A homemade photoporation platform, mediated by gold nanoparticles (AuNPs), was constructed to find a suitable method for finding all adherent cells in this process with high delivery efficiency. The thermal dynamics of AuNPs could be monitored. Based on this system, 60 nm AuNPs were selected to be attached to cells for optimal photoporation. After irradiating the cells covered with AuNPs using a nanosecond pulse laser, fluorescein isothiocyanate-dextran in the medium were delivered into optoporated adherent HeLa (human cervical cell lines) cells. The delivery efficiency and cell viability of this process were evaluated using a fluorescence microscope and flow cytometry. The experimental results showed that targeting cells using antibodies, laser irradiation from the top of the cell culture well, and reducing the cell medium are important for improving the delivery efficiency. The optimal loading efficiency for adherent HeLa cells was 53.4%.

## 1. Introduction

The successful treatment of serious diseases (e.g., cancer) based on gene therapy requires the transport of foreign materials into cells. Many cell lines derived from humans and animals are adherent cells [1,2]. Accordingly, there is a need to enable the delivery of foreign materials into adherent cells without disturbing them.

Several delivery approaches have been developed to date, e.g., biological, chemical, and physical methods, including microinjection, electroporation, magnetoporation, and sonoporation [3,4,5,6,7]. However, effectively transferring foreign materials into adherent cells using current approaches presents several challenges. These techniques have drawbacks in terms of completing transfection involving adherent cells. For example, when using chemical methods, chemical toxicity and chemical vectors require an escape from endosomal compartments [8,9]. In biological methods, genotoxicity and immune responses [5,10] require many specifically designed electrodes for electroporation [11,12,13] and the agglomeration of magnetoporation after the removal of the magnetic field for magnetoporation [7]. Furthermore, ultrasonic cavitation is rarely precision-controlled for sonoporation [7].

Photoporation is an alternative physical technique for transferring adherent cells. Based on this technology, Lalonde et al. [14] used a 532-nm nanosecond pulsed laser to treat adherent cells (MW278); the results indicated a maximum 25% delivery efficiency at 50 mJ/cm^2^. Schneckenburger et al. [15] reported delivering green fluorescence protein into adherent cells (CHO-K1) using a 488-nm argon-ion laser with less than 30% transfection efficiency. The delivery efficiency in these studies was low for adherent cells.

Recently, photoporation mediated by gold nanoparticles (AuNPs) has received increasing attention. When irradiating AuNPs using short laser pulses (<10 ns), their surface temperature rapidly increases, and the surrounding water evaporates, resulting in vapor nanobubbles (VNBs) around the AuNPs [16,17,18]. These VNBs can induce the formation of shockwaves and micro-jets, resulting in the delivery of molecules [19]. A small number of studies have reported that a nanosecond pulsed laser inducing VNBs in mediated photoporation can also achieve a high transfection efficiency for adherent cells [20,21]. However, the adherent cells used in these works were detached from the culture dish and suspended in a culture medium for experimental purposes. Our previous study found that the optimal delivery efficiency for adherent cells was only 30%, while in the case of trypsinized cells, this value increased up to 70% [22]. Although delivery efficiency improved, trypsinized treatment disturbed membrane protein and affected the cytoskeletal system. Also, the trypsin-treated cells were not biologically stable, which may have adversely affected their recovery [23,24].

Studies have also focused on designing a plasmonic substrate that could be used to seed adherent cells, and when excited by a nanosecond laser, it improves delivery efficiency. For example, Mazur et al. [25] seeded adherent human cervical cell lines (HeLa) cells on a self-fabricated plasmonic substrate activated by a nanosecond pulsed laser to form transient pores in a cell membrane for cargo delivery. This method resulted in a delivery efficiency of 95% for 0.623-kDa-sized membrane-impermeable fluorescent dye while maintaining 98% cell viability. In a recent study [26], a new plasmonic substrate, a self-assembled thermoplasmonic titanium nanocavity, was designed on which adherent HeLa cells were incubated. This platform was used to deliver calcein green (0.623 kDa) into cells at an efficiency of 78% and viability of 87%. Although the delivery efficiency in these studies was high, it was closely related to the molecular weight of foreign materials. False-positive results may occur if the molecular weight is too small. Therefore, it is difficult to provide a reference for the delivery of macromolecular materials.

Wu et al. [27] reported a silicon-based delivery chip on which they seeded cells. It was able to transfer larger items such as bacteria, antibodies, enzymes, and nanoparticles into adherent cells such as HeLa, normal human dermal fibroblasts, and renal proximal tubule epithelial cells with high efficiency and cell viability. These research results demonstrate good delivery efficiency for photoporation. However, a significant flaw is that they represent non-selective properties for transferring cells and are not suitable for in vivo transfection. Unlike the studies noted above, Xiong et al. [19] reported an excellent photoporation system for delivering macromolecules into adherent cells. No other research, however, has shown comparably similar high delivery efficiencies.

In the current study, we established a homemade photoporation setup to deliver foreign material into adherent cells. The AuNPs were used as mediators to assist photoporation, and a 532-nm pulsed laser was employed to irradiate cells from their upper surface that the AuNPs attached to. We found that the delivery efficiency of naked AuNP- mediated photoporation was low for adherent cells. Accordingly, the AuNPs were modified using epidermal growth factor receptor (EGFR) antibodies, which could attach more stably to the cell membrane. The results indicated that functionalized AuNPs could significantly improve the delivery efficiency of adherent cells compared to naked AuNPs. In addition, we further improved the delivery efficiency of adherent cells by reducing the volume of the cell culture medium, which absorbs the laser energy. As a result, the delivery efficiency reached 53.4% for 10-kDa fluorescein isothiocyanate (Fitc)-dextran. Accordingly, functionalized AuNP-mediated photoporation is a process that can be used to develop drug delivery into adherent cells.

## 2. Materials and Method

### 2.1. Homemade Photoporation Platform

The optical layout of the designed photoporation platform is shown in Figure 1. A pulse-laser beam (wavelength: 532 nm, pulse duration: 6 ns, repetition frequency: 10 Hz) was emitted using a Q-switched neodymium-doped yttrium aluminum garnet pulsed laser system (Q-smart 450, Quantel, Cournon d’Auvergn, France) and focused on the cell membrane by a 10-fold objective lens. A λ/9 beam plate split the laser beam, and an energy meter (PE50-DIF-C; Ophir, Jerusalem, Israel) recorded the laser energy. An XY motion stage carried a 96-well plate, which was moved line-by-line using a step size of 100 μm. The imaging light-path comprised a white light source, and an electron-multiplying charge-coupled device camera was used to adjust and ensure that the focal spot acted on the cell.

### 2.2. Cell Culture

The HeLa cells were obtained from the cell bank of the Chinese Academy of Sciences (Shanghai). Cells were grown in 100-mm glass-bottom dishes and cultured in Dulbecco’s modified Eagle’s medium (Hyclone, Logan, UT, USA) and supplemented with 10% fetal bovine serum (Gibico, Grand Island, NY, USA) and 1% penicillin/streptomycin (Solarbio, Beijing, China) at 37 °C in a 5% CO_2_ atmosphere. Cells were detached by trypsin (Hyclone, USA) and reseeded in a 96-well plate one day before the experiment.

### 2.3. Antibody and Polyethylene Glycol Modification of Gold Nanoparticles

For modifying AuNPs with antibody, thiol-PEG-carboxyl (SH-PEG-COOH) (MW: 5000 Da; Xi’an ruixi Biological Technology Co., Ltd., Xi’an, China) was used to conjugate antibodies to AuNPs. First, 400-μL SH-PEG-COOH (20 μM) was mixed with 10 μL N-(3-dimethylaminopropyl)-*N*-ethylcarbodiimide hydrochloride (EDC, Sigma-Aldrich, St. Louis, MO, USA, 0.4 mg/mL) and 10-μL *N*-hydroxy succinimide (NHS, Sigma-Aldrich, USA, 1.1 mg/mL) to react under agitation at room temperature for 4 h. Then, 5 μL of the EGFR antibody (1F4; Cell Signaling Technology, Danvers, MA, USA, 2 μg/μL) was added to the mixture to react at room temperature for 2 h. Following that a 400-μL AuNP solution (60 nm, 2.6 × 10^10^/mL) was added to the reaction product and reacted at 4 °C for 4 h. Afterward, it was centrifuged (8000 rpm, 15 min) to remove the by-product, resuspended in ultrapure water, and stored at 4 °C for use within three days.

For modifying AuNPs with polyethylene glycol (PEG), 400 μL SH-PEG-COOH (20 μM) was mixed with 400 μL AuNPs solution (60 nm, 2.6 × 10^10^/mL) and reacted at 4 °C for 4 h. After that, the conjugation was centrifuged (8000 rpm, 15 min) to remove the unreacted PEG and resuspended in the same volume of ultrapure. Then, 10 μL EDC and 10 μL NHS were added into the mixture to react under agitation at room temperature for 0.5 h. Afterward, it was centrifuged (8000 rpm, 15 min) to remove the by-product, resuspended in ultrapure water, and stored at 4 °C for use within three days.

An ultraviolet–visible spectrophotometer (UV–VIS; V-550, JASCO, Tokyo, Japan) was used to collect the absorption spectra of the naked AuNPs and their constructs. The spectral interval was set as 1 nm, and spectral range was set from 350 nm to 750 nm. The zeta potentials and particle sizes were also measured using a Zetasizer Nano ZSE (Malvern, Malvern, UK). Three samples were measured to obtain the zeta potentials and particle sizes. The type of zeta potential sample pool was DTS 1060. The laser source posited on the back of sample pool. For stability assessment, the AuNPS, PEG-coated AuNPs and antibody@PEG coated AuNPs solutions were stored for 12 and 24 h and illuminated by a red laser beam to observe its dispersion state.

### 2.4. Confirmation of the Attachment of Gold Nanoparticles on the Cell Membrane

The cells were incubated with the naked AuNPs for 1 h and with the modified AuNPs for 0.5 h at a cell-to-AuNPs ratio of 1: 2000. Afterward, they were washed with phosphate-buffered saline (PBS). Following that, 2.5% glutaraldehyde was added into the petri dish and stored at 4 °C for one night. Next, different concentrations of ethyl alcohol (30%, 50%, 70%, 80%, 90%, 95%, and 100%) were sequentially used to dehydrate the cells from a low to a high concentration. Finally, the cells were dried at 50 °C, and 5-nm AuNPs were sprayed onto the surface of the cell membranes for measurement using a scanning electron microscope (SEM; GeminiSEM 500, Zeiss, Oberkochen, Germany). The transmission electron microscopy (TEM) photograph of AuNPs was measured by a field emission TEM (JEM-F200, JEOL, Tokyo, Japan). Image J was used to calculate the size distribution from TEM images.

### 2.5. Cell Photoporation Experiments

Before conducting the photoporation experiments, HeLa cells were seeded in a 96- well plate with a density of 0.8 × 10^4^ cells per well. Before conducting the laser treatment, HeLa cells were incubated with AuNPs for 1 h and with modified AuNPs for 30 min. Then, the cells were washed with PBS to remove the non-adhered AuNPs. Afterward, a fresh culture medium containing the foreign materials of Fitc-dextran (10 kDa; Xi’an ruixi Biological Technology Co., Ltd., Xi’an, China) was added to yield a final concentration of 0.4 mg/mL.

Next, the homemade platform was used to treat the cells. After completing the treatment, cells were placed in a humidified incubator for 30 min. Then, propidium iodide (PI; Solarbio, Beijing, China) was added to yield a final concentration of 1 μg/mL after washing cells with PBS. Ten minutes later, cells were washed with PBS and supplied with a fresh culture medium. Afterward, the fluorescence images of the prepared cell samples were collected using a fluorescence microscope (Eclipse Ti; Nikon, Tokyo, Japan).

### 2.6. Measurement of Delivery Efficiency

The cells were also collected to quantify cell viability and delivery efficiency using flow cytometry (FACScan; BD Biosciences, San Diego, CA, USA). The cell samples were detached by trypsin (Hyclone, Logan, UT, USA), resuspended in a culture medium, and measured using a flow cytometer. A minimum of 5000 cells was collected for the analysis.

## 3. Results

### 3.1. Conjugation and the Characterization of Antibody-Modified Gold Nanoparticles

To ensure that more AuNPs were attached to the cell membrane, SH-PEG-COOH was used to conjugate the AuNPs and EGFR antibodies (see Figure 2a). The characterization of AuNPs and their constructs are shown in Figure 2b–d, and Table 1 shows the statistical results. The naked AuNPs were measured with a plasmon absorption peak at 536 nm, an average potential of −33.1 mv and an average diameter of 60.6 nm. The PEG-coated AuNPs were measured with a plasmon absorption peak at 542 nm, an average potential of −9.2 mv and an average diameter of 112.3 nm. The antibody@PEG-coated AuNPs were measured with a plasmon absorption peak at 541 nm, an average potential of −23.7 mv and an average diameter of 217.5 nm. The UV–VIS spectra showed that the plasmon absorption peak of the antibodies modified by the AuNPs shifted from 536 to 541 nm (Figure 2b). In addition, the conjugation size was characterized by dynamic light scattering (DLS). The pristine AuNPs had a diameter of approximately 60.6 nm, while the diameters of the PEG-coated AuNPs and antibody@PEG-coated AuNPs were 112.3 and 217.5 nm, respectively. Furthermore, the average potentials of PEG-coated AuNPs and antibody@PEG-coated AuNPs had changed to −9.2 and −23.7 mv, respectively. These results demonstrated that EGFR antibodies had been successfully conjugated on the AuNPs. For the stability assessment, the AuNPs, PEG-coated AuNPs and antibody@PEG-coated AuNPs solutions were stored for 24 h and the dispersion state was observed at different time points. It showed good dispersion without aggregation for 24 h (Figure 3). The results suggested that AuNPs and their conjugations have good stability for photoporation, which can be done in a few hours. We next investigated the adherent cells’ photoporation mediated by functionalized AuNPs.

### 3.2. Confirmation of the Attachment of Gold Nanoparticles on the Cell Membrane

Figure 4a shows the TEM image of AuNPs and their size distribution based on TEM images. To confirm the attachment of AuNPs on the cell membrane, SEM images of HeLa cells were taken following their incubation with naked AuNPs for 1 h (Figure 4c,e), and with antibody-modified AuNPs for 0.5 h (Figure 4d,f) at a cell-to-AuNPs ratio of 1:2000. The distribution of AuNPs on the cell membrane was observably localized in the SEM images. Compared with the naked AuNPs, more antibody-modified AuNPs can be observed on the cell membrane in Figure 4d and the zoomed-in image (Figure 4f). This demonstrated that AuNPs were present on the cell membrane following incubation. Accordingly, we further explored the photoporation of adherent cells.

### 3.3. Photoporation Mediated by Naked Gold Nanoparticles

To verify the performance of the photoporation platform for adherent cells, we first treated the HeLa cells with different laser fluence settings. A 10-kDa Fitc-dextran (FD10) was used as an extracellular material to evaluate the delivery efficiency and cell viability. Following incubation with AuNPs at a cell-to-AuNPs ratio of 1:2000, the cells were treated using two different laser fluence settings. The PI was added to the cell to characterize the cell viability. Figure 5 shows the fluorescence images of the intracellular delivery of FD10 for the adherent HeLa cell. A group treated with a high laser fluence (6.4 J/cm^2^) showed a high delivery efficiency compared to a low laser fluence-treated group (1.28 J/cm^2^), but was accompanied by a high mortality. The bright-field images showed that the high laser fluence-treated cells were deformed. Thus, we systematically evaluated the delivery efficiency of FD10 and the cell viability for different laser fluences. After completing the laser treatment, cells were incubated for 30 min to ensure the death of late apoptotic cells and to avoid affecting the delivery efficiency. Afterward, PI was added to the cells’ culture medium to characterize the cell viability. In Figure 6, Q1 represents perforated cells, Q2 and Q3 represent dead cells, and Q4 represents intact viable cells. Table 2 shows the statistical results for the perforated and dead cells. They indicate that there was no significant increase in the delivery efficiency when the laser fluence was less than or equal to 2.56 J/cm^2^. As the laser fluence increases, delivery efficiency shows an obvious enhancement up to 39.4% at 6.4 J/cm^2^. However, the percentage of dead cells also sharply increases to 37.8%. Accordingly, it is difficult to achieve a high delivery efficiency with low mortality using naked AuNP-mediated photoporation for adherent cells. We posit that this may be because of the limited amount of AuNPs rendered adherent on the cell membrane by adhesive attraction. Thus, we modified the AuNPs using an antibody to promote their attachment to the cell membrane.

### 3.4. Improvement of the Delivery Efficiency

Functionalized AuNPs were used to assist in the photoporation to achieve a high delivery efficiency with low mortality at a low laser fluence. The cell-to-AuNPs ratio was set as 1:20,000. The laser fluence threshold of the bubble generation was decreased by increasing the number of AuNPs attached to the cell membrane. In this way, we attempted to deliver the FD10 into HeLa cells and obtained a high delivery efficiency at a low laser fluence (see Figure 7). The group of functionalized AuNPs (Group D) showed obvious increases in delivery efficiency, which went up to 20.2% at 1.28 J/cm^2^, while the group of non-specifically bound AuNPs (Group C) achieved only 4.3%. The mortality of the group of functionalized AuNPs was only 8.56%. Accordingly, we concluded that functionalizing AuNPs was a good approach for improving the delivery efficiency of photoporation for adherent cells.

Laser energy was absorbed and scattered by the cell culture medium. To further improve the delivery efficiency for adherent cells, the cell culture medium volume was reduced. The results (Figure 7) showed that the delivery efficiency of the 50-μL medium (Group F) increased to 53.4%, which was significantly higher than the 70 μL medium (Group E); no obvious decrease in the cell viability was observed. The fluorescence images indicated the same results. It is noted that in the bright-field images, cells did not show obvious morphological changes. These results indicated that the delivery efficiency had been significantly improved for adherent cells.

## 4. Discussion

It is important to find a versatile extracellular material delivery method that is suitable for cells. Although the NP-mediated photoporation method is suitable for both floating and adherent cells, it requires improving the delivery efficiency for adherent cells [22].

In our previous work on the adherent cell as a model for studying molecule delivery to adherent cells, we found that the optical delivery efficiency was approximately 30% with a 20% death rate. A higher nanoparticle concentration and a higher fluence induced a death rate greater than 50% [22]. In such a case, we irradiated cells with focused light from the bottom of the 96-well plate. Nanosecond laser pulses with high fluence focusing on a solid target ionized the target surface and led to high-density plasma formation, accompanied by shockwave emission during plasma expansion. The shockwave emission strongly relied on the interface attached to the solid target. For example, when the target was covered with transparent materials (glass), the shockwave strength and duration ranged from 5 to 10-fold and 2 to 3-fold, respectively, which were higher than direction ablation [28]. It is reasonable to assume that the higher death rate of cells was due to the shockwave generation.

In the following experiments, the laser illuminated samples from the top of the cell plate without a cover. The delivery efficiency was affected by the concentration of AuNPs [22]. The AuNPs that had been functionalized with antibodies toward the cell surface markers showed enhanced accretion on the cell membranes [29]. We modified the AuNPs using mixed PEG and EGFR antibodies. SEM characterized the attachment of AuNPs to the cell membrane. Additional AuNPs were observed following their modification with antibodies. These results showed that the delivery efficiency significantly increased with the assistance of functionalized AuNPs (to 20.2%), compared to the 4.33% of naked AuNPs, even with a low irradiation fluence of 1.28 J/cm^2^. However, in our previous work we found that death rate increased with the increase in the concentration of AuNPs [30], and when the concentration of AuNPs was lower than a certain amount, the viability of cells was hardly affected by the irradiation fluence [31], which is confirmed in this work, as shown in Figure 6 and Figure 7. As shown in Figure 6 (Table 2), when the lower concentration was used, we found that the loading efficiency increased with an increase in irradiation fluence, but the death rate was almost constant. When the concentration of AuNPs increased to 1:20,000 (cell-to-AuNPs ratio) from 1:2000, both the death rate and the loading efficiency increased with the increase in irradiation fluence. Moreover, antibody-conjugated AuNPs tend to form clusters through antibody-mediated endocytosis [32], which makes it easier to induce cell death.

A higher irradiation fluence of 2.56 J/cm^2^ was used to improve the delivery efficiency, as shown in Figure 7. Both the delivery efficiency and death rate were enhanced, the former showed an approximate increase ranging from 20% to 30%, while the latter experienced an approximate 8% increase to 18%. From the above results, we deduced that higher irradiation fluence should induce a higher degree of cell death. We found both the perforated cells and dead cells increased especially after binding AuNPs on cell membrane. Considering the dead cells as a factor, we observed that group F showed a more significant increase in perforated cells than that of group E, while dead cells hardly increased. So, we inferred that group F was the optimal parameter to obtain true efficiency in this study.

In all experiments, as shown in Figure 5 and Figure 7a, we found that some cells did not show any fluorescence after treatment with PI and Fitc, which implied these were intact viable cells. This phenomenon is mainly attributed to the probability of bubble generation. In this work, the cell perforation was induced by VNBs, while VNBs generation is an event with a nondeterministic probability [18], which resulted in part of intact viable cells being available.

Since the surrounding media of the cells influence cell membrane permeability [22], the volume of the culture medium around the treated cells was decreased from 70 to 50 μL in the experiments; this included sufficient nutrition for the treated cells and a fluence of 2.56 J/cm^2^ was used to irradiate the sample. We found that the delivery efficiency was significantly improved (53.4%) and was accompanied by a lower cell death rate of 21%, as shown in Figure 7. One reason for this could be less attenuation of laser energy by less cell culture medium, which included some light absorbing materials, such as not cleared gold nano-particles. This could provide relatively higher irradiation fluence, which is suitable for the optimal parameters.

To improve the delivery efficiency, we changed the way that pulsed laser irradiated the cells in the homemade photoporation platform. We irradiated the cell from above the cell, which prevents the impact of shockwaves on cell viability caused by irradiation from the bottom of the cell culture dish. On the other hand, antibody was used to modify AuNPs, and hence, more AuNPs attached on the cell membrane; moreover, the antibody-conjugated AuNPs could form clusters, which can significantly improve the delivery efficiency at low irradiation fluence due to the reduction in the bubble generation threshold [32].

## 5. Conclusions

In conclusion, our work reported a homemade platform for delivering extracellular materials into adherent cells based on NP-mediated photoporation. We irradiated the cell from above the cell to prevent the impact of shockwaves on cell viability caused by irradiation from the bottom of the cell culture dish. EGFR antibodies modified the AuNPs to create additional AuNPs that could attach to cell membranes and decreased the threshold fluence of VNB generation. Assisted by functionalized AuNPs, the delivery efficiency increased to 20.2% at 1.28 J/cm^2^, while this value was only 4.3% when assisted by naked AuNPs. This demonstrated that the delivery efficiency can be improved by functionalized AuNPs at low irradiation fluence. Furthermore, the delivery efficiency was improved by reducing the medium volume, which reached 53.4%. This study serves as a reference for delivering extracellular materials into adherent cells.

## Figures and Tables

**Figure 1 membranes-11-00550-f001:**
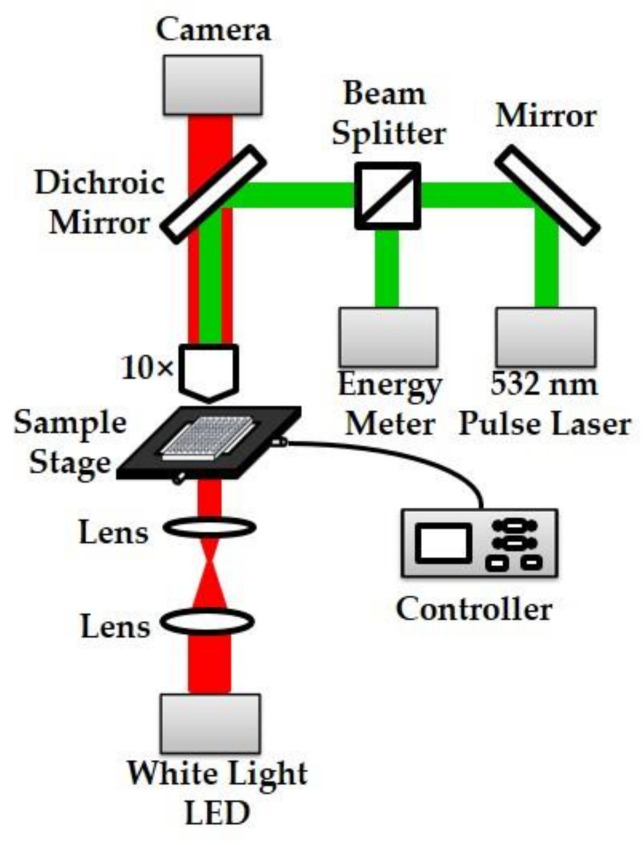
Optical layout of the designed photoporation platform.

**Figure 2 membranes-11-00550-f002:**
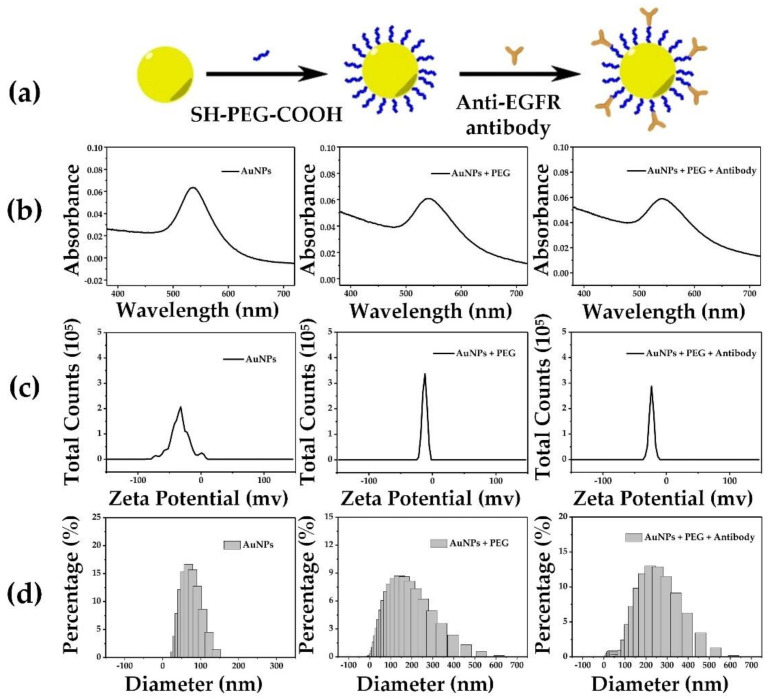
Design and characterization of EGFR antibody-modified AuNPs. Left panel: the AuNPs; middle panel: the AuNPs@PEG complex; right panel: the AuNPs@PEG@antibody complex. (**a**) A schematic presentation of AuNPs’ bioconjugation; (**b**) the UV–VIS spectra of naked AuNPs, AuNPs@PEG, and AuNPs@PEG@antibody; (**c**) the zeta potentials; (**d**) the particle diameter measurements at the different stages of AuNP coatings.

**Figure 3 membranes-11-00550-f003:**
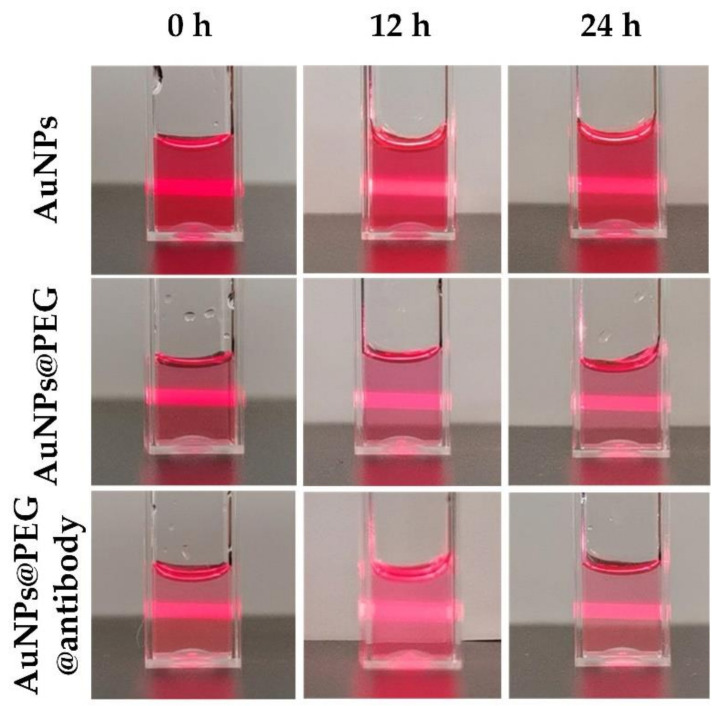
Photos of AuNPs, PEG-coated AuNPs and antibody@PEG-coated AuNPS solution illuminated by a laser beam at different time points.

**Figure 4 membranes-11-00550-f004:**
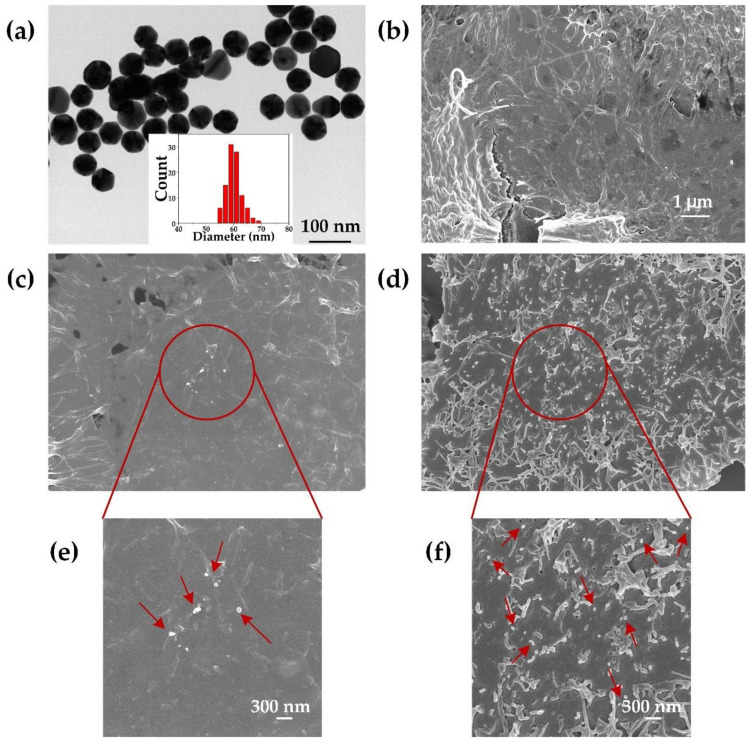
(**a**) The TEM photograph and a graph (insert graph) with size distribution of AuNPs from TEM images. (**b**) SEM images of HeLa cells without incubation with AuNPs. (**c**) SEM images of HeLa cells incubated with naked AuNPs (60 nm) and an incubation time of 1 h. (**d**) SEM images of HeLa cells incubated with antibody-modified AuNPs at an incubation time of 0.5 h; (**e**,**f**) are the zoomed-in images.

**Figure 5 membranes-11-00550-f005:**
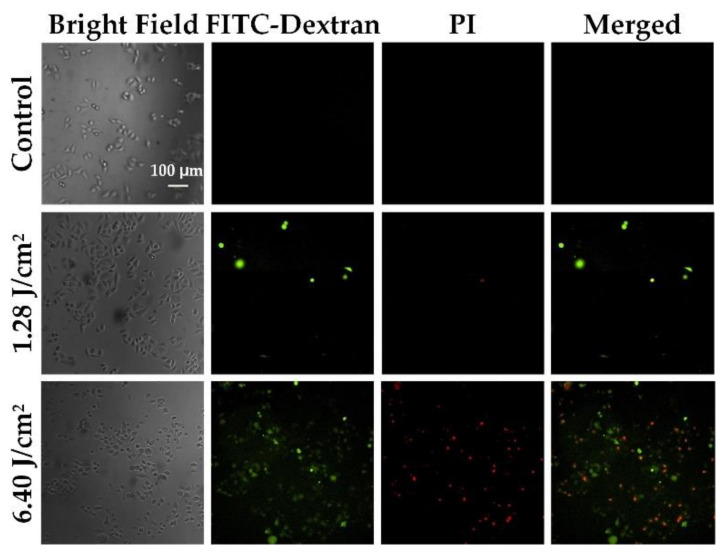
Fluorescence images of the intracellular delivery of FD10 for HeLa cells after treatment with different laser fluences.

**Figure 6 membranes-11-00550-f006:**
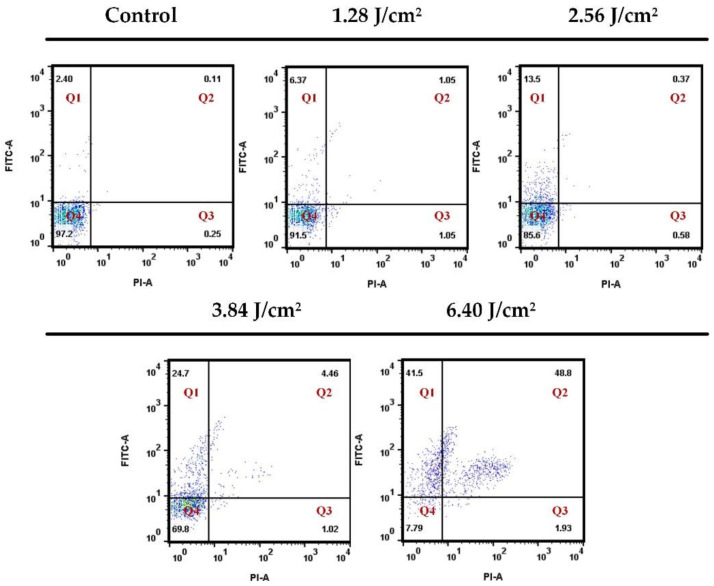
Percentage of perforated cells (Q1), dead cells (Q2 and Q3), and intact viable cells (Q4) determined by flow cytometry for HeLa cells at different laser fluences. Q1, Q2, Q3 and Q4 represented Fitc positive and PI negative cells, Fitc positive and PI positive cells, Fitc negative and PI positive cells and Fitc negative and PI negative cells, respectively.

**Figure 7 membranes-11-00550-f007:**
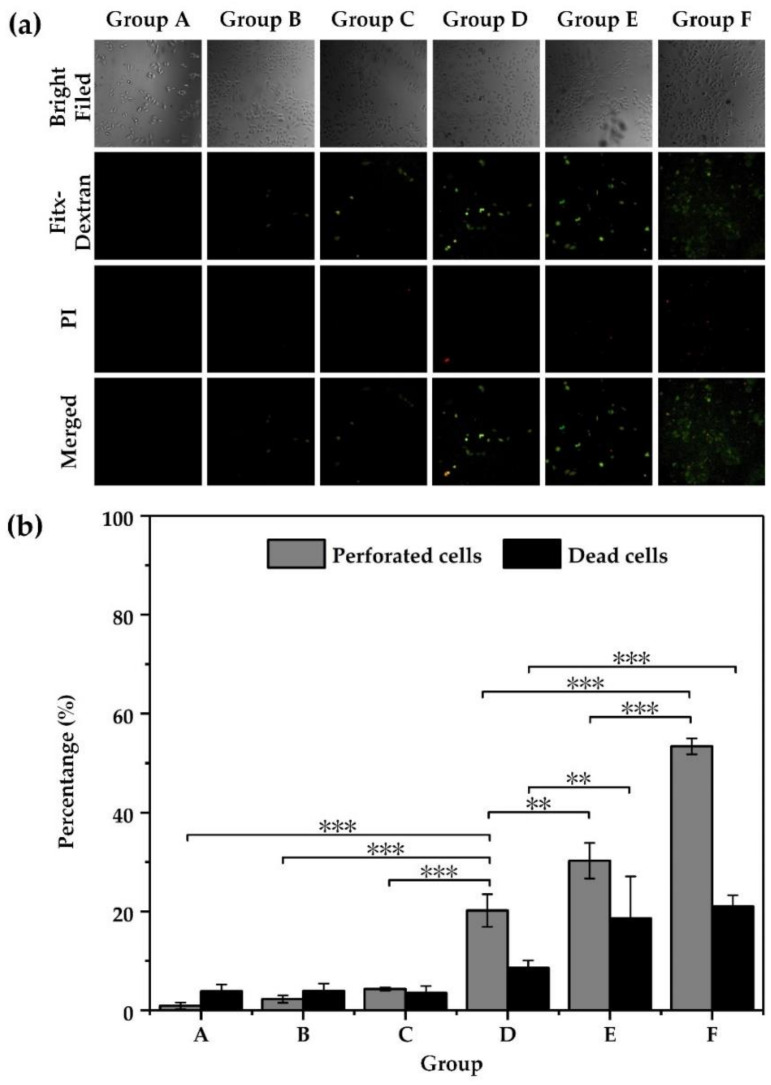
Delivery of Fitc-dextran (10 kDa) into HeLa cells by EGFR antibody functionalized AuNPs’ mediated photoporation. (**a**) Fluorescence images of the intracellular delivery of FD10 for HeLa cells after treatment with laser-irradiated antibody-modified AuNPs. (**b**) The percentage of perforated and dead cells determined by flow cytometry for HeLa cells. Statistically significant differences are indicated by ** (*p* < 0.01) and *** (*p* < 0.001). Group A (with naked AuNPs): laser fluence: 0 J/cm^2^; culture medium: 70 μL. Group B (with antibody modified AuNPs): laser fluence: 0 J/cm^2^; culture medium: 70 μL. Group C (with naked AuNPs): laser fluence: 1.28 J/cm^2^; culture medium: 70 μL. Group D (with antibody modified AuNPs): laser fluence: 1.28 J/cm^2^; culture medium: 70 μL. Group E (with antibody modified AuNPs): laser fluence: 2.56 J/cm^2^; culture medium: 70 μL. Group F (with antibody modified AuNPs): laser fluence: 2.56 J/cm^2^; culture medium: 50 μL.

**Table 1 membranes-11-00550-t001:** Characterization of AuNPs and their constructs.

	Peak Position (nm)	Zeta Potential (mv)	Diameter (nm)
AuNPs	536	−33.1 ± 2.0	60.6 ± 0.5
AuNPs@PEG	542	−9.2 ± 1.3	112.3 ± 1.1
AuNPs@PEG@Antibody	541	−23.7 ± 0.2	217.5 ± 4.7

**Table 2 membranes-11-00550-t002:** Statistical results for perforated and dead cells.

	Control (%)	1.28 J/cm^2^ (%)	2.56 J/cm^2^ (%)	3.84 J/cm^2^ (%)	6.4 J/cm^2^ (%)
Perforated Cells	2.36 ± 1.43	6.36 ± 1.57	11.6 ± 2.51	22.9 ± 5.14	39.4 ± 5.55
Dead Cells	0.45 ± 0.18	1.08 ± 0.73	1.07 ± 0.4	6.57 ± 2.12	37.83 ± 9.33

## Data Availability

Not applicable.

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
