# Peer review of "Delivery of Foreign Materials into Adherent Cells by Gold Nanoparticle-Mediated Photoporation"

_membranes, 2021, doi:10.3390/membranes11080550_

Round 1
Reviewer 1 Report
The authors applied a homemade photoporation platform as well as functional AuNP to increase delivery efficiency into cells.
Comment 1: why when 2.56 J/cm3 shows higher perforated cells and similar dead cells compared to 1.28 J/cm3 in figure 5 and table 2, but figure 6b shows both higher perforated cells and dead cells?
Comment 2: For data in Figure 4&6., the authors need to explain why when cells are treated with PI and FTIC dyes, they didn't show any fluorescence intensity.
Comment 3: When calculated with high delivery efficiency, the authors also need to include the death cells as factors to demonstrate the true efficiency.
Comment 4: For the improved delivery efficiency, the authors need to elaborate the impacts made by photoporation platform and functional AuNP, respectively, to demonstrate the true novelty of the methods.
Reviewer 2 Report
The paper entitled:"Delivery of Foreign Materials into Adherent Cell by Gold Nanoparticles Mediated Photoporation" from Xiaofan Du et. al. has been revised. The manuscript contains an interesting work about the synthesis and characterization of 60 nm core size gold nanoparticles mixed with PEG and modified with EGFR antibody for drug delivery applications mediated by photoporation. Moreover, a homemade photoporation platform has been developed by the authors. The viability of the nanosystems in Hela cells was carried out and the loading efficiency it was of 53.4% in Hela cells.The paper is well-written and in general the conclusions are well-supported by the experimental data. In my opinion the paper is worth to be published in Membranes journal after minor revision.
-No information regarding stability of AuNPs and mixed with PEG and modified with EGFR antibody nanoparticles are reported. This information is needed to discard possible aggregation phenomena which could interfere in medical applications of the developed nanosystems.
-A TEM photograph and a graph with size distribution of AuNPs from TEM images are needed to verify the core size reported by the authors.
-Abbreviations should be used and defined after they were firstly defined in text: SH-PEG-COOH, EGFR antibody, Hela cells...
-More information about some experimental technique used in this work is needed. More specifically about UV-visible and zeta potential technique. For instance, what is the wavelenght intervale of measurement, the spectral bandwidth or the interval of measurement. Respect the zeta potential experiments the number of repetitions should be clarified, the type of cubete and laser position.
-On page 4, line 156, a comment about the satbility of AuNPs based on its charge and size seems to be pertinent. The same could be applied for the other nanosystems described in table 1.
Reviewer 3 Report
- Section 3.1. Lines 158-169. Please indicate the method which was applied to determine the nanoparticle size.
- Section 2.3 describes the conjugation of sh-peg to antibody by carbodiimide method and after this adsorption of modified product to AuNPs. But the results shows the presence of nanoparticles modified with peg-antibody and modified with peg. Please add the method of conjugation of nanoparticles with sh-peg without antibody to the section 2.
- Why did not you use the conjugate of nanoparticles with antibody without peg? It will be very interesting to compare the results obtained with two types of conjugation -covalent and physical adsorption.
Reviewer 4 Report
I find the paper very interesting and reporting on an important topic, with important results. Although the paper is understandable, in terms of English, it would still benefit from moderate english changes. Some examples:
Abstract:
60 nm gold nanoparticles (AuNPs) were selected to attached to cells (better write "to be attached")
Experiment results (Experimental results)
Introduction
therapy of the serious diseases (therapy of serious diseases)
Many cell lines derived from humans and animals are adherent cell (adherent cells)
Therefore, it is of great meaningful (Therefore, it is very meaningful)
deliver foreign materials into adherent cell (adherent cells)
etc
The help of a native engliush speaker of someone with equivalent knowledge would be benefitial.
Round 2
Reviewer 1 Report
I recommend to accept the article.
Reviewer 3 Report
The authors very carefully and scrupulously reviewed the manuscript, made the appropriate changes to it. In this form, it can be accepted for publication.